# NoFumo+: A Clinical Trial of an mHealth for Smoking Cessation with Hospitalized Patients

**DOI:** 10.3390/ijerph181910476

**Published:** 2021-10-06

**Authors:** Patricia García-Pazo, Albert Sesé, Jordi Llabrés, Joana Fornés-Vives

**Affiliations:** 1Department of Nursing and Physiotherapy, University of the Balearic Islands, Balearic Island, 07122 Palma, Spain; Joana.fornes@uib.es; 2Balearic Islands Health Research Institute (IdISBa), Hospital Universitario Son Espases, Balearic Island, 07120 Palma, Spain; albert.sese@uib.es (A.S.); jordi.llabres@uib.cat (J.L.); 3Department of Psychology, University of the Balearic Islands, Balearic Island, 07122 Palma, Spain

**Keywords:** smoking behavior, mHealth, cognitive behavioral therapy, clinical trial

## Abstract

Smartphone applications (apps) improve accessibility to smoking cessation treatments. The NoFumo+ app administers a cognitive behavioral therapy program for smoking cessation. This study evaluates the efficacy of NoFumo+ for quitting smoking or reducing cigarette consumption versus the usual information-based treatment. A clinical trial was conducted with 99 hospitalized smokers, 54 pseudo-randomly assigned to the app treatment and 45 to the usual treatment. The two groups had homogeneous baseline characteristics to ensure comparability. Abstinence was evaluated at post-treatment (two months) and at a six-month follow-up. The results obtained indicate that participants who receive the usual treatment are 5.40 times more likely to continue smoking than those who undergo the app treatment (95% CI = [1.35; 20.15]). Participants who do not succeed in quitting smoking with the app manage to decrease their habitual consumption. Users who successfully complete treatment with NoFumo+ access all its contents and use the chat, but without requesting professional support. There is not enough empirical evidence to attribute this success to any specific element of the app. NoFumo+ achieves better abstinence rates than the usual information-based treatments, and the goal of generalizing its use to the non-hospitalized smoking population may be achievable in the future.

## 1. Introduction

Smoking is associated with more than 25 types of pathologies, and it continues to be one of the main causes of preventable death in the world [1]. If the current smoking level continues, in 10 years more than seven million smokers worldwide will die from smoking [2]. Moreover, it is necessary to add the social and healthcare costs that a country’s smoking population may incur, due to direct costs stemming from the disease itself (such as hospitalization, among others) and indirect costs related to loss of productivity [3].

Seventy percent of smokers want to quit smoking, and 50–60% try to quit but without any professional help [4]; therefore, the long-term success rate is low, from 3 to 5% [5]. However, there are numerous smoking cessation interventions cited in the United States clinical practice guidelines (USCPG) that have been shown to be effective [6]. These interventions range from short-term brief advice to behavioral therapy (BT), such as cognitive behavioral therapy (CBT), alone or combined with pharmacotherapy (varenicline, bupropion, and nicotine replacement therapy (NRT)) in cases of greater dependence [7]. The clinical practice guidelines (CPG) of the Spanish Society of Pneumology and Thoracic Surgery (SEPAR) recommends that healthcare personnel take advantage of any hospitalization of smokers to offer them some kind of smoking cessation treatment [8], given that success rates of over 40% have been obtained [9,10]. Unfortunately, most healthcare services do not currently follow these recommendations, either because the professionals are not trained in this area or because they do not have time to perform this activity, among other reasons [11].

It is a public health priority to make effective programs more accessible to the entire smoking population [12]. For this purpose, different intervention formats have been developed through informative leaflets, motivational messages, emails, “Quitlines”, or Internet programs, among others, achieving greater improvements in abstinence rates when they are more adapted to the users [13,14]. Currently, with the emergence of new information and communication technologies (ICTs), smartphone health applications, or mHealth, offer personalized and interactive interventions that provide relevant remote support for smoking cessation [15]. In addition, studies have shown that smokers express greater interest in receiving smoking cessation treatments through ICTs than through other methods [16], even while the smoker is hospitalized [17].

However, most of the existing mHealth apps on the market were not developed following the recommendations of the USCPG, given that they do not incorporate the brief advice known as the mnemonic rule of the 5 A’s: Ask, Advise, Assess, Assist, and Arrange [6]. Nor do they include BT [18] such as CBT [19], even though scientific evidence indicates that their inclusion improves efficacy rates [20]. Instead, some apps include acceptance and commitment therapy (ACT) [19], which is not cited in the CPGs [6,7,21]. Moreover, publications about these apps do not describe how they implement behavior change techniques or specify which techniques they use, making it difficult to study and compare them with other similar apps [14]. 

Addressing these previous shortcomings means including the following elements in the development of new mHealth: creating a cessation plan with goals and objectives [20], monitoring smoking behavior, providing continuous feedback to the user, sending personalized motivational messages [14], and incorporating evidence-based behavior change techniques [22], cognitive-behavioral techniques, and long-term follow-ups and social support [23]. Moreover, there is a need for more research with randomized or pseudo-randomized trials that provide consistent evidence [24].

Consequently, this research team developed an mHealth called NoFumo+ app, which includes all the elements that are described above and missing in other apps. It also follows the SEPAR CPG recommendations specifically for the hospitalization period [8]. Thus, the evidence shows the effectiveness of a high-intensity intervention that takes place for at least four weeks after discharge from the hospital [25,26]. NoFumo+ app was piloted by experts and users, and it was rated as a simple, intuitive, and optimal app for the goal of quitting smoking.

The main objective of this study is to evaluate the efficacy of the NoFumo+ app for smoking cessation or reducing cigarette consumption, compared to the usual treatment (information brochure), in hospitalized smokers. An efficacy evaluation is performed at two months post-treatment and a follow-up at six months post-treatment. The second objective is to describe the use of mHealth by smokers who complete the NoFumo+ app treatment and remain abstinent at the post-treatment control. 

## 2. Materials and Methods

### 2.1. Design

A two-branch, parallel, longitudinal, pseudo-randomized clinical trial was conducted with an experimental group (EG) that received treatment using the NoFumo+ app and a control group (CG) that received treatment as usual (TAU). The effect of the treatment was monitored two months after it began (post-treatment) and 6 months after hospital discharge, both by telephone. The study was approved by the ethics committee of the Balearic Islands (protocol code IB 3865/19) and was registered in ClinicalTrials.gov with reference NCT04502004.

### 2.2. Participants

The following inclusion criteria were used: (1) age over 18 years old; (2) smoking five or more cigarettes per day; (3) expressing a desire to remain abstinent after hospital discharge; (4) a minimum hospitalization period of 24 h in the reference hospital of the study; (5) having a personal Smartphone (Android or IOs); (6) understanding the Spanish language; (7) having sufficient cognitive capacity to understand the treatment. Exclusion criteria were: (1) being diagnosed with any mental illness, (2) substance abuse, or active alcoholism, and (3) participating in a smoking cessation treatment prior to admission. The initial sample was composed by smokers admitted to a public hospital in the Migjorn health sector in the Balearic Islands. It was obtained between November 2019 and April 2020.

### 2.3. Intervention

#### 2.3.1. NoFumo+

The EG received treatment through the NoFumo+ app, available free of charge for Android and iOS devices, after obtaining an access code provided by the hospital’s Tobacco Detoxification Unit (TDU). The app administered a multicomponent program [27,28] with CBT, consisting of second-generation CT with extensive scientific evidence in the field of smoking cessation [29]. This therapy aims to help the smoker identify triggers for smoking behavior and develop coping strategies to achieve abstinence from smoking. In addition to cognitive-behavioral techniques, other elements offered in these programs are motivational techniques (e.g., through daily messages), promotion of alternative healthy activities, coping skills training, and relapse prevention. Moreover, the app added the 5A’s recommended by the USCPG, it was adapted to the hospital context, and it offered psychoeducation about effective pharmacological treatments. The treatment lasted 30 days and was distributed in 15 sessions represented by numbered boxes that formed a circle (Table 1).

Every three sessions made up a treatment phase. The circle was gradually completed as participants finished the phases. Smokers accessed the content sequentially, unblocking activities as they acquired knowledge and progressed through the treatment (Figure 1a). Knowledge acquisition was assessed by means of three multiple-choice questions after each phase, and the user could advance in the treatment if he/she answered two of them correctly. To encourage adherence to treatment, the app employed daily notifications as reminders as well as gamification elements (e.g., collection of medals, an avatar, etc.). NoFumo+ app monitored smoking behavior through self-reporting, daily or whenever the user wished, recording: the number of cigarettes smoked, the situations in which they smoke, their desire to smoke, and their emotional state. Pharmacological treatment could also be monitored if the participant was receiving it. Participants were provided with feedback on their self-reporting by means of a graph, which allowed them to visualize their evolution and identify critical situations for smoking (Figure 1b). NoFumo+ app had an emergency button for moments of stronger craving. This function offered guidelines to avoid relapses, the possibility of contacting the TDU by email, and the connection to online games as distraction elements.

#### 2.3.2. Treatment-As-Usual (TAU)

The treatment received by the CG consisted of an information brochure developed by the study hospital that included the SEPAR “Decalogue for smoking cessation”. It was delivered in printed format. The contents of the brochure included: (1) guidelines to prepare for the cessation phase (e.g., selection of the day when they are not going to smoke); (2) action plan (e.g., thinking of alternatives to smoking); and (3) guidelines to remain abstinent (e.g., promoting physical activity). In addition, they were advised to seek more support from community resources that can intervene in smoking cessation, and some benefits of quitting smoking were cited, such as improved smell, taste, and skin, as well as decreased risk of diseases caused by tobacco. The information brochure can be found in Appendix A (Table A1).

### 2.4. Procedure

First, the recruitment process was carried out by the medical staff who were admitting patients to the hospitalization units. After identifying smokers by consulting their clinical history and confirming that they smoked, staff members questioned them about their interest in giving up smoking. If they were interested, the hospital’s TDU was contacted. Second, within 48 h, a member of the research team invited the smoker to participate in the study. After signing the informed consent, an initial assessment was made of the patient as a potential participant in the study by means of an interview and psychometric questionnaires, which are described in the measures section. Third, after ruling out the subjects who met the exclusion criteria, the participants were pseudo-randomly distributed in each of the two branches of the trial (EG and CG), so that participants were assigned to each group alternately. Finally, prior to discharge, the EG participants received access codes to download the NoFumo+ app and technical assistance in using the application, whereas the CG participants received the informative brochure.

### 2.5. Measures

#### 2.5.1. Initial Interview

Information was collected through a semi-structured interview about hospital admission (e.g., admission unit, reason for hospitalization), sociodemographic variables (e.g., age, sex), smoking history data (e.g., daily use, age of onset of use), and whether or not they were receiving pharmacological treatment (e.g., NRT or varenicline).

#### 2.5.2. Instruments 

Information was gathered on specific smoking and emotional variables. The Fageström Test [30] was used to assess the degree of nicotine dependence, and the Motivational Richmond Test [31] was used to assess motivation to quit smoking. The emotional questionnaires used were the State-Trait Anxiety Questionnaire (STAI-E/R) [32], the Beck Depression Inventory-II (BDI-II) [33], and the State-Trait Anger Expression Inventory (STAXI-E/R) [34]. All the questionnaires are internationally known and have shown evidence of reliability and validity.

#### 2.5.3. The App Interface

It collected users’ interactions with the different elements of the app, allowing access and monitoring by TDU staff. Every time the user accessed a function of the app, it was recorded, along with the minimum time spent in the app. This last measure is conservative because it did not count the time spent connected to Internet links the app provided. Some of the data collected by the interface included: the number of days the app was used from the time it was downloaded until the treatment was completed; the use of the chat, based on the number of days connected and time spent in it; and the use of the emergency button, such as the number of times it was accessed. In addition, the interface also distinguished between sending an email to the TDU and the use of online games.

#### 2.5.4. Control and Follow-Up Interviews

Information was collected through two telephone interviews about self-reported abstinence or cigarette consumption (control and follow-up), as well as the degree of satisfaction with the app in the EG. This non-objective assessment of abstinence is considered valid for remote interventions with little contact with the professional, given that a low incidence of false testimony has been found [35]. Regarding cigarette consumption in people who failed to remain abstinent, it was important to measure it because a reduction in consumption is associated with a greater likelihood of quitting [36]. The degree of satisfaction with the app was recorded using an analog scale (0–100). Likewise, the elements of the application that were more or less useful were also detected.

### 2.6. Data Analysis

The test data matrix was processed and cleaned to avoid the presence of errors. Descriptive statistics and normality tests (Shapiro–Wilk) were used to study the behavior of the trial variables. Non-parametric tests were used to cope with the normality assumption non-compliance due to the small sample and to avoid potential outliers’ effect. To check the initial equivalence and comparability between the control and experimental groups, tests of comparison of proportions were carried out in the case of categorical variables, and non-parametric median comparison tests for quantitative variables, all of them for independent samples.

Likewise, to evaluate the efficacy of the NoFumo+ app, the difference between the percentage of smokers and nonsmokers in the CG and EG groups at post-treatment and follow-up was analyzed using a chi-square test (intention-to-treat analysis). This analysis was also carried out considering only the participants in the EG who completed the app treatment, referred to as EG2 (per-protocol analysis). Subsequently, to aid interpretation, the relative risk was analyzed using odds ratios and their 95% confidence intervals. As a measure of efficacy, we also analyzed in parallel whether there were statistically significant differences between the number of cigarettes consumed by the participants in the two groups who failed to quit smoking in the pre-treatment stage, at the post-treatment control, and at the six-month follow-up. To do this, non-parametric median comparison tests were used. Finally, a comparative analysis of the emotional measures at post-treatment and follow-up was also performed. Non-parametric median comparisons tests were applied for the last two cases.

Regarding the analysis of app use, the second objective of the trial, given that only two subjects continued to smoke despite having completed the treatment, it was not possible to analyze which app-specific elements (gamification, social chat, videos, etc.) predict therapeutic success. Nevertheless, a descriptive analysis of use was carried out of the variables that reflected the users’ interaction with the app in the group of participants who finished the treatment, specifically smokers who remained abstinent at post-treatment. All statistical analyses were conducted with the SPSS 27.0 program [37].

## 3. Results

The initial sample was composed of 110 smokers admitted to a public hospital in the Migjorn health sector in the Balearic Islands. After applying the inclusion-exclusion criteria mentioned above, the study sample consisted of 99 smokers: 45 in the CG and 54 in the EG. At the end of the treatment, the size of the CG had declined to 35 participants at post-treatment and 34 subjects at the six-month follow-up. The EG was reduced to 50 smokers at post-treatment and 38 at follow-up (Figure 2).

Regarding the comparability of the pre-treatment groups, they presented homogeneous baseline characteristics, with no statistically significant differences in any of the variables of interest (Table 2). The questionnaires showed (Fageström test and the motivational Richmond test) that both groups had moderate nicotine dependence and were highly motivated to stop smoking. The results of the questionnaires at the beginning of the treatment indicated that the medians of the participants presented low levels of anxiety and depression and the absence of anger when the intervention began.

### 3.1. Efficacy of the App at Post-Treatment and Follow-Up

The results on the efficacy of the NoFumo+ app show that, after completing the treatment, the probability of smoking is 50% for both the CG (17/35) and the EG (32/50) as intention-to-treat analysis; that is, there is no statistically significant difference between the two groups (*p* = 0.18). Moreover, there are no significant differences between receiving one or the other treatment at six-month follow-up, although 16 participants in the CG and 24 in the EG remain abstinent (*p* = 0.12). However, taking into account only those subjects who completed treatment with the app, the per-protocol analysis, a group called EG2, 42% of the EG (21/50), 90.5% (19/21) do not smoke two months after starting the treatment, compared to 48% who do not smoke in the CG (*p* = 0.002). This difference is also observed at six months, given that 88.9% (16/18) of the EG2 do not smoke, compared to 44% (15/34) of the CG (*p* = 0.002). The odds ratio value obtained according to the risk analysis for completing treatment with the app is less than 1 (*OR* = 0.09; *95% CI* = [0.02; 0.49]); therefore, the probability of smoking in the CG is 5.40 higher than in the EG2 (*RR* = 5.40, *CI95%* = [1.39; 20.97]).

We also analyzed the number of cigarettes smoked by users who failed to quit smoking in the CG and the EG. In the CG, the median number of cigarettes smoked at baseline is 21 (*Q_1_–Q_3_* = 15.0–30.0), 12 (*Q_1_–Q_3_* = 2–15) at post-treatment (2 months), and 14 (*Q_1_–Q_3_* = 10–20) at follow-up. In the EG, the median at baseline is 13 cigarettes (*Q_1_–Q_3_* = 9–20), 7 (*Q_1_–Q_3_* = 5–10) at post-treatment, and 7 (*Q_1_–Q_3_* = 5–10) at follow-up. These differences are statistically significant at six-month follow-up (*p* < 0.045). Finally, with regard to the emotional variables recorded, no statistically significant differences are obtained between the two groups in the study, or between the participants who remained abstinent in the CG and the EG2, at post-treatment and at follow-up. (Table 3).

### 3.2. Adherence Using the App

The data the interface provides indicate that 61% (33/54) of the EG do not complete the app treatment, so that 38.8% (21/54) drop out in the first phase, 18.5% (10/54) in the second phase, 3.7% (2/54) the third phase, and 0% (0/54) in the fourth phase. Therefore, 38.8% (21/54) reach the fifth phase, thus ending the treatment.

Participants who do not smoke after completing the app intervention are in treatment for a median of 42 days (*Q_1_–Q_3_* = 30.0–56.0). These participants use the social chat on a median of 16 days during treatment (*Q_1_–Q_3_* = 3.7–24.5), and in this time they make a median of 19 connections, (*Q_1_–Q_3_* = 3.25–54.25), spending an average of 21.6 min in the chat (*Q_1_–Q_3_* = 4.7–70.5). In addition, during the treatment, they use the emergency button on a median of 15 days (*Q_1_–Q_3_* = 7.2–40.0), they do not send emails to the TDU service the entire time they use the application (*Q_1_–Q_3_* = 0–1.2), and they do not connect to the online games (Table 4).

Regarding the degree of satisfaction with the app, on a scale from 0 to 100, the mean score for users who finish the treatment is 80.5 (SD = 24.8). The elements rated positively are: self-reporting, the informative videos by professionals, and the chat. Among the comments recorded, the NoFumo+ app is rated as a simple, motivating mHealth that has quality information and serves as a distraction in times of crisis.

## 4. Discussion

This clinical trial aims to evaluate the efficacy of the NoFumo+ app for smoking cessation, which implements the USCPG recommendations and includes CBT, compared to treatment as usual (information brochure), in hospitalized smokers. The results obtained show that smokers who receive treatment as usual are 5.40 times more likely to smoke after hospital discharge than those who receive the NoFumo+ app treatment. Completing treatment with the NoFumo+ app leads to a 90.5% probability of abstinence. Likewise, people who do not quit smoking, either because they do not complete the mHealth treatment or because they are not successful after completing it, and smoke during follow-up, smoke less than their habitual consumption.

To date, very few mHealth apps implement both interventions (USCPG and CBT) and rigorously evaluate their efficacy. Moreover, we have not found any that take place during the user’s hospitalization. Therefore, it is difficult to compare the results obtained in this trial with those of other similar apps. Some of these apps are Quit Genius [38], which implements only CBT, and Smart Quit [19], which adds ACT elements to CBT in its first version. The publications about its functioning are qualitative [39] and descriptive [19], and only two studies evaluate the effectiveness of the Smart Quit app. The first study compares the Smart Quit app to the Quit Guide app, which only implements USCPG, and its results indicate an abstinence rate of 13% versus 8%, respectively, at the 60-day follow-up [40]. The second study compares the Smart Quit app to an updated version of itself (Smart Quit app 2.0) that uses ACT. In this case, an efficacy of 11% is observed for the new version, compared to 13% for the old version, at the two-month follow-up. Moreover, the abstinence rate of the Smart Quit app 2.0 increases to 28% when only considering participants who complete the treatment [41]. The literature agrees that apps that implement CBT achieve better dropout rates than those that only apply USCPG recommendations [38,42]. In addition, users who do not remain abstinent significantly reduce their habitual consumption, as in Smart Quit 2.0 [41], Quit Genius [38], and NoFumo+ applications.

The results of the present trial suggest that by combining USCPG recommendations and CBT into one app, NoFumo+ obtains better abstinence rates in a six-month follow-up period. In addition, users of apps that implement the CBT report being more satisfied and motivated to quit smoking than with mHealth that only incorporates USPGC recommendations [39]. These results coincide with the assessment made by NoFumo+ users, who value the self-monitoring of smoking behavior as a motivating element.

In our study with NoFumo+ and in Smart Quit 2.0 applications, the abstinence rate is higher in users who finish the treatment offered by the app, although with NoFumo+, 42% finish the treatment (21/50), whereas with Smart Quit 2.0, only 24% do so (24/99). This could be due to the fact that the NoFumo+ app resolves some of the shortcomings of the Smart Quit 2.0 app by incorporating gamification elements, evaluating comprehension of content through questionnaires, and unblocking content based on the user’s progress [41].

Unlike most mHealth applications, the NoFumo+ app has a sequential operation, so that the user accesses different treatment phases while progressing in the acquisition of knowledge about techniques of this treatment. This way of operating makes it possible to more reliably verify that users have completed the treatment and, thus, establish the relationship between treatment completion and the abstinence rate [40].

Apps that do not present a sequential treatment associate better abstinence rates with the number of times users interact with the application. In this regard, the Smart Quit app (first version) is more effective, with an interaction rate of 37.2 times, versus the Quit Guide app, with a rate of 15.2 times [41]. These two ways of assessing effectiveness, relating abstinence to treatment completion or to the frequency of interaction, limit comparisons of different apps [40]. 

In the case of the NoFumo+ app, participants who finish their treatment and remain abstinent at a six-month follow-up make use of the application chat. The literature associates the availability of a within-app chat with greater treatment effectiveness [43]. The chat fulfills a function of social support (reading conversations of other users, sharing achievements) and distraction in moments of craving [44]. We cannot specify the intentions of NoFumo+ participants when using it, although we observed that other elements incorporated in the app with the same function, such as online games, were not employed by these users. Another function that was underused by abstinent users was the e-mail communication with hospital professionals. This result coincides with what was obtained in other mHealth [16], as well as with Quitline, with only 3–4% of smokers taking advantage of it [14], even though as many as 48% of users requested this service within the apps when surveyed [16].

Finally, this study presents a different sample composition from those in the mHealth efficacy studies mentioned above with regard to age and recruitment method. In the case of age, NoFumo+ app users are older (m = 53 years) than in the other studies cited (m = 38–41 years). In a study of predictors of the use of the Smart Quit app, older age is associated with greater adherence to the app [45]. Moreover, there is a noteworthy difference in the recruitment method because our trial recruits hospitalized patients with pre-existing pathologies, and the other studies generally involve random samples of participants recruited through social networks. The literature points out that hospitalization of the smoker is a good learning moment [4], given that it is usually associated with greater perceived vulnerability and the smoker may be in a motivational phase described as “ready for action or action” [42]. It is likely that for this reason we obtained better dropout rates in both treatments (EG and CG) in this study. Most relapses in these patients occur in the days following hospital discharge [46,47], which coincides with the high dropout rate from app treatment in the first week after leaving the hospital. Thus, during this time, contact with users should be increased, and the most attractive activities should be offered. 

The study limitations are mainly related to the small sample size. Taking into account that not all the participants in the EG completed the treatment, and that of those who did, only two did not quit smoking, it was not possible to carry out comparative analyses to assess the elements that facilitate adherence to the app and potential predictor variables of the efficacy of the NoFumo+ app. For the same reason, the lack of statistical power to detect significant differences may underlie the lack of change in the emotional variables evaluated. Therefore, it is necessary to carry out new clinical trials with the NoFumo+ app using larger more heterogeneous samples from different contexts, in order to perform predictive analyses of its effects on both the emotional variables and the elements that make up the treatment and test its efficacy in the general smoking population. Technological advances, and specifically mHealth, are a priority in order to achieve improvements in health and, therefore, reduce the costs of public health systems.

## 5. Conclusions

The new NoFumo+ app was able to increase the abstinence rate, compared to conventional treatment, after the smoker was discharged from the hospital. Although we did not obtain sufficient evidence to attribute the app’s success to any of its specific elements, we can say that what is important is the rigorous implementation of CBT and CPG recommendations in this mHealth app, along with the elements of gamification, chat, etc. The efficacy obtained in this trial raises hopes about the therapeutic possibilities of the NoFumo+ app, which, as a tool for mass use, could contribute in later versions to analyzing which of the apps’ functions best support therapies to quit smoking.

## Figures and Tables

**Figure 1 ijerph-18-10476-f001:**
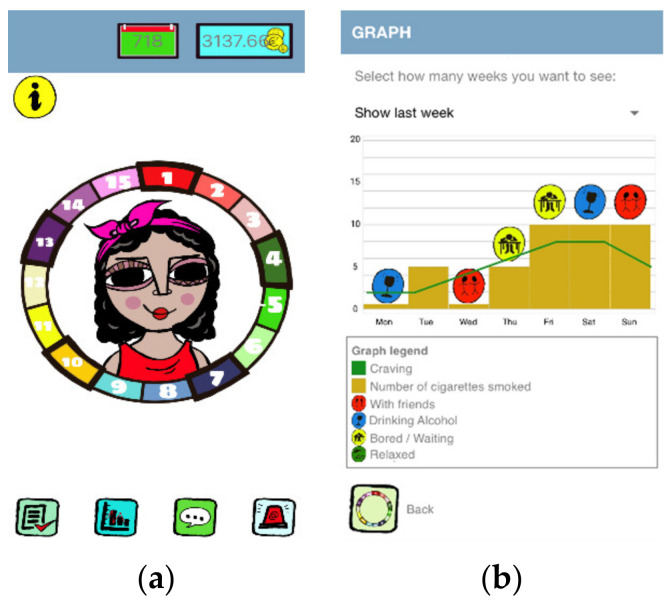
These figures show screenshot of NoFumo+ app: (**a**) main screen with treatment completed; (**b**) graphical resources describing smoking behavior analysis.

**Figure 2 ijerph-18-10476-f002:**
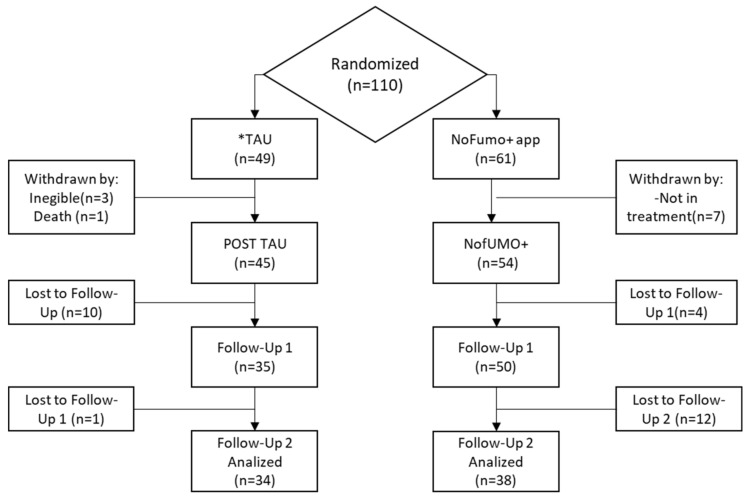
Participant flow diagram. Treatment-As-Usual (*TAU).

**Table 1 ijerph-18-10476-t001:** NoFumo+ app multicomponent program.

Box	Information (Text/Audio)	Contents (Short Videos and Leaflets)	Activities (Interactive and Notes)	Links (Internet/Web Pages)
1	Treatment plan (Establish objectives). Stimulus control (“Golden Rules”).	Tobacco components; nicotine; controlled breathing.	Relaxation training; encouraging social support (friend/family); recording benefits for those who quit smoking now.	Recommendations to stop smoking.
2	The analysis of smoking behavior. Prevention of response. Advice about change of routine.	Health effects of tobacco, pharmacological advice.	Write down reasons to stop smoking, encourage social support.	Information on how tobacco affects health.
3	Behavior analysis; the desire to smoke. Contents: Activities.	Abstinence syndrome. Advise on coping strategies.	Select and/or note optional activities.	Alternative coping strategies for withdrawal symptoms.
4	Maximize the self-rewarding experience. Contents	Physical benefits of quitting smoking. Social support. Distraction. Prevention of response.	Activities: Training in coping skills for risk situations.	Benefits of quitting smoking.
5	Analyze smoking behavior and desire to smoke, personal reasons to quit.	Identify risk situations.	Relaxation (Jacobson and autogenous). Feedback graphics.	Controlled breathing, self-guided relaxation technique (Jacobson).
6	Remember to control stimuli that remind you to smoke (e.g., situations, smoke-free spaces …).	Effective pharmacological treatments to stop smoking.	Activities to maintain abstinence.	Life without tobacco.
7	History of tobacco. Raising awareness in tobacco advertising. Assert yourself as a non-smoker.	Informing about the history of tobacco, psychological part of addiction.	Alternative activities for risk situations; Social support.	Harmful advertising about smoking.
8	Time management. Graphical feedback of behavior analysis.	Social support (chat). Distraction.	Smoking behavior analysis charts.	Common myths.
9	Negative thoughts.	Advice about errors in thinking and their modification; social chat.	Write down negative thoughts and modify them; talk to yourself.	Change in thinking.
10	Stress management.	The problem-solving technique.	Technical problem-solving training.	Problem-solving technique.
11	Advise about social skills as a technique.	Talking to oneself: Thought-stopping technique; social skills for cigarette refusal.	Training in social skills.	
12	Reinforcing abstinence as a reward and objective of the program.	Cognitive restructuring: True story (COPD).	Action in the event of a fall or/and relapse.	The reason for the fall and relapse.
13	Strengthen the ex-smoker’s identity (smoking is not an option).	The urge to smoke, the temptation.	Testimony of 2 ex-smoking patients (1 year and 25 years of abstinence).	Resources/testimonies from other ex-smokers.
14	Reinforcing alternative smoking behaviors.	Physical exercise of different intensities. Psychoeducation and behavioral activation.	Use a pedometer.	Program to promote physical exercise at different ages.
15	Reinforcement and re-objectives of the treatment in terms of monitoring in person and through the application.	Weight control. Training in planning (diet and exercise).	Harvard plate.	Healthy eating education and recipes.

Chronic Obstructive Pulmonary Disease (COPD).

**Table 2 ijerph-18-10476-t002:** Baseline characteristics of both control and experimental groups, and comparison tests significance *p*-value.

Characteristics		Control Group	Experimental Group	*p*-Value
*n* = 99		45.5% (45)	53.5% (54)	
Reason for admission	Respiratory	48.8% (21)	54.7% (29)	0.55
	Cardiovascular	18.6% (8)	20.8% (11)	1.00
	Other	32.6% (14)	24.1% (13)	0.50
Demographics	Age (*Med, Q_1_–Q_3_*)	59.0, 48–67	54.0, 42.7–63.2	0.29
	Woman	37.7% (17)	25.9% (14)	0.69
	Man	62.2% (28)	55% (30)	0.62
	Married	53.5% (23)	51.9% (28)	0.84
Educative level	Primary studies or less	50.0% (21)	46.0% (23)	0.77
	Secondary studies	09.5% (4)	12.0% (6)	1.00
	Bachelor	14.3% (6)	12.0% (6)	0.77
	Vocational training	16.7% (7)	18.0% (9)	0.79
	University	09.5% (4)	12.0% (6)	0.63
Tobacco History	Smoking family; Yes	47.5% (19)	48.0% (24)	0.62
	Consumption start age	15.0, 13.0–19.0	16.0, 14.0–18.0	0.40
	Previous attempts; No	78.6% (33)	84.6% (44)	0.21
	Cigarettes per day	21.5, 13.5–30.0	20.0, 10.0–20.0	0.45
	Smoking years (*Med, Q_1_–Q_3_*)	42.0, 34.0–51.0	40.0, 26.0–45.0	0.40
	Cumulative dose/year (*Med, Q_1_–Q_3_*)	33.0, 17.5–63.0	32.1, 12.3–45.2	0.62
	Co-oximetry (CO) (*Med, Q_1_–Q_3_*)	3.0, 2.0–6.0	4.0, 1–41	0.37
	Smoking family, Yes	47.5% (19)	48.0% (24)	0.62
Pharmacotherapy	None	69.0% (20)	67.3% (35)	0.93
	TSN ^1^	13.8% (4)	11.5% (6)	0.89
	Vareniclina	17.2% (5)	21.2% (11)	0.72
Questionnaires (*Med, Q_1_–Q_3_*)	T. Fageström ^2^	4.00, 3–6	5, 3.25–6	0.30
	T.Richmond ^3^	7.50, 6–9.25	7.0, 5–9	0.30
	BDI-II ^4^	9.00, 5–20.50	11.0, 6–16.50	0.55
	STAI ^5^: State	19.0, 10.75–31.0	21.0, 12.5–30.0	0.77
	Traits	21.0, 15.0–31.75	18.0, 13.0–29.0	0.25
	STAXI ^6^: State	10.0, 10–12	10.0, 10–12	0.99
	Traits	15.0, 12.0, 19.0	14.0, 12.0–18.0	0.89

*Med* = median; *Q_1_* = quartile 25%; *Q_3_* = quartile 75%; unless otherwise specified, data are presented as % (*n*); CO = carbon monoxide; ^1^ Nicotine patches; ^2^ Fagerström Test for Nicotine Dependence; ^3^ Richmond Test Motivation; ^4^ Beck Depression Inventory (BDI-II); ^5^ State-Trait Anxiety Inventory (STAI); ^6^ State-Trait Anger Expression Inventory (STAXI); *p*-values < 0.05; Fisher’s exact and median comparison test.

**Table 3 ijerph-18-10476-t003:** Median comparisons between post-treatment and following non-smokers from GC and GE2.

	Questionnaires	GC *Median* (*Q_1_–Q_3_*)	GE2 *Median* (*Q_1_–Q_3_*)	*p*-Value *
Post-treatment 2 months	BDI-II ^1^	11.0 (11.0–11.5)	7.0 (3.5–15.25)	0.68
	STAI ^2^-Estate	12.5 (7.2–19.5)	8.5 (2.5–22.25)	0.68
	STAXI ^3^-Estate	10.0 (10.0–10.0)	10.0 (10.0–10.75)	0.62
Following 6 months	BDI-II ^1^	11.0 (6.00–13.5)	4.0 (3.0–14.0)	0.37
	STAI ^2^-Estate	14.5 (8.70–30.5)	14.0 (4.0–19.0)	0.64
	STAXI ^3^-Estate	11.0 (10.0–12.7)	10.0 (10.0–10.0)	0.79

GC = control group; GE2 = experimental group (completed treatment), *Q_1_* = quartile 25%; *Q_3_* = quartile 75%; ^1^ Beck Depression Inventory II (BDI-II); ^2^ State-Anxiety Inventory (STAI); ^3^ State-Anger Expression Inventory (STAXI). * *p*-values < 0.05; median comparison test.

**Table 4 ijerph-18-10476-t004:** Adherence GE2 non-smoking participants post-treatment.

Engagement	GE2 *Med* (*Q_1_–Q_3_*)
Days used	42.00 (30.00–56.00)
Chat Interactions	19.00 (3.25–54.25)
Days chat	16.00 (3.70–24.50)
Chat duration (minutes)	21.65 (4.70–70.50)
Emergency Interactions	15.00 (7.20–40.00)
Send e-mail (frequency)	0.00 (0.00–1.20)
Games online	0.00 (0.00–0.00)

*Med* = Median; *Q_1_* = quartile 25%; *Q_3_* = quartile 75%.

## Data Availability

The data presented in this study are available on request from the corresponding author. The data are not publicly available due to patients’ privacy.

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
