# Peer review of "NoFumo+: A Clinical Trial of an mHealth for Smoking Cessation with Hospitalized Patients"

_ijerph, 2021, doi:10.3390/ijerph181910476_

Round 1
Reviewer 1 Report
This a study that assesses the effects of “NoFumo+” app by conducting a pseudo-randomised clinical trial. The results show that using the app achieved better abstinence rates than the usual treatment. The authors have not listed the lines in the text, so it is difficult to specify the corrections to be done.
Methods
- Although the authors have followed the Consort guide, it is important for authors to adapt the text according to the specifications mentioned in the guide:
- Regarding participants, only election criteria and origin of participants must be presented
- The flow chart and baseline characteristics must be in the results section, according to the structure of the CONSORT guide.
- The main response variables are not very clear, that is, those that allow predicting the success of the intervention
- In the statistical analysis authors must make clear when tests are used for independent samples and when for paired samples.
Results
- Authors must present the flow of participants and the table of baseline characteristics in the first place, as required by the CONSORT guide.
- Authors have to specify whether EG was used to perform an intention-to-treat analysis. It looks like that apparently a protocol analysis was performed for EG2, since the presentation of the analysis is confusing. In clinical trials, an intention-to-treat analysis is recommended, while by protocol is not. Then, the results of both analyzes must be specified, because the authors only present the analysis by protocol.
- Generally, when the medians are presented, instead of the standard deviation, the (First quartile - Third quartile) are presented. Please correct the results of the first paragraph, page 9.
- At the footnote of each table authors need to specify the statistical procedure used.
Reviewer 2 Report
First of all, I congratulate the authors for the research. It remains essential to treat smokers and obtain the highest possible smoking cessation rate.
Observations:
1. Review figure 1, and especially the number of people lost to follow-up and final follow-up.
2. It would be of interest to add a table with the data related to the efficacy of the app at post-treatment and follow-up. This would add clarity to these results, which are the most important part.
3. Check the orthography of some references, errors are detected.
